# Climatic Variability in Altitude: Architecture, Thermal Comfort, and Safety along the Facade of a Residential Tower in the Mediterranean Climate

**Soultana (Tanya) Saroglou [1,\*], Hofit Itzhak-Ben-Shalom [2,3] and Isaac A. Meir [1]**

[1] Department of Civil and Environmental Engineering, Faculty of Engineering Sciences, Ben-Gurion University of the Negev, Beer-Sheva 8410501, Israel; sakis@bgu.ac.il

[2] Department of Architecture, Sami Shamoon College of Engineering (SCE), Beer-Sheva 84100, Israel; hofitit@sce.ac.il

[3] Department of Environmental Studies, The Porter School of the Environment and Earth Sciences, Tel Aviv University, Tel Aviv 52621, Israel

\* Correspondence: saroglou@post.bgu.ac.il

**Abstract:** This research is part of a wider study on the sustainability of high-rise buildings within the urban environment. The focus here is on wind and temperature stratifications per different height through in situ monitoring on a high-rise residential tower in the Mediterranean climate of Tel Aviv, and their impact on thermal comfort and user safety. The appropriateness of design is discussed in relation to the direct exposure to higher-up wind velocities, the thermal perception of the residents through questionnaires, and the safety and usability of the outdoors space according to height. The potential for advancing the energy efficiency of the structure is also discussed. The study covers a hot and a cold season, focusing on the specificities of the wind regime in the specific climate, through seasonal variations. Results from the monitoring of data confirmed increased wind and gust velocities per building height all year round, reaching the level of danger for the occupants during winter, cancelling, thus a successful operation of the outdoor balcony space. The occupants' perception and use of the outdoor balcony space per building height were in direct relationship to the increased wind velocities per height. Discussion and conclusions critically evaluate the residential high-rise building typology in the Mediterranean climate through the design of the outdoor balcony space along the height of the envelope. The results set an initial understanding and delineation for future studies, while underlining the complications of designing and occupying tall buildings and the level of detailing required.

**Keywords:** residential high-rise design; wind climate per building height; thermal perception per building height; Mediterranean climate; seasonal variations

## 1. Introduction

The numbers of high-rise buildings around the world have been growing rapidly over the last decades, with the 21st century portraying increasingly vertical urban environments. However, while the high-rise typology is embraced in many of the world's cities, the performance of such buildings, in relation to sustainable strategies, is lacking the scientific background that it requires. One such consideration relates to the high-energy consumption in high-rise buildings in comparison to all other construction [1–3] and the need to comply with strict energy regulations around the world [4–6]. Building energy efficiency has become especially challenging in today's energy intensive era, while most of the high-rise building stock is mechanically supported, consuming large amounts of energy, relative to the buildings' increased volume and scale. As far as 'green credentials' are concerned, the percentage of buildings that pursue this procedure and aim towards improving the tall-building performance is growing; however, these numbers are still small on a global scale, and mainly refer to office environments [7–13].

However, 'building sustainability' is a wide term, and it does not necessarily mean that projects that employ a number of sustainable strategies or solutions or present a number of sustainability credentials are on the whole successful [2,14], with the actual energy needs of the structure after completion being an important parameter. In addition, advanced green technologies have not yet yielded the success they were meant to have, while their realization can potentially skyrocket a project's budget, e.g., the installation of wind turbines for electricity production. At the same time, regulations like the 2010/31/EU Energy Performance of Buildings Directive (EPBD) state that all new buildings must be nearly Zero Energy Buildings (nZEBs) by 2020 [4]. The implementation of passive strategies towards the achievement of this goal is a must, i.e., natural ventilative cooling in reducing energy loads. The success of natural ventilation is relative to building orientation, seasonal variations [15,16], and building height, while its implementation also maintains a thermally comfortable, healthy indoor environment [13,17].

Nevertheless, designing a high-performance high-rise building that incorporates renewable energy sources (RES) and non-renewable ones is a complex matter, that requires a much higher flexibility on budget and time [8,18]. The Council of Tall Buildings and Urban Habitat (CTBUH) defines a building of 14 or more floors, or more than 50 m, being within the threshold of the 'tall building' definition [19]. But, while the focus on sustainable tall-buildings mainly points towards iconic architecture of technological and structural innovation in the height range of 300–400 m [20], an increasing range of 50–100 m high buildings is designed and built without green credentials, certification, or any form of prior studies through simulation and modelling. So, despite an increase in building height, these higher-than-average buildings most likely do not satisfy the requirements of sustainable construction and lack studies relating to their successful operation, while design guidelines for tall buildings are generally lacking.

Whereas research on appropriate strategies to bridge the gap between the high-rise buildings and the new era of energy codes is vital [21,22], further issues that arise with the increase in building height within the urban fabric are equally pressing and concern social, environmental, economic factors, and more. In this process, site-specific studies in neighborhoods and urban areas become important in order to start mapping the interconnections between environmental variables and urban morphology [23–27]. The topic of urban winds and high-rise buildings, for example, has gained popularity in recent years being of growing relevance to the increasing complexity of this building typology within the urban environment [28–35]. In addition, urban winds and high-rise buildings have been studied from an urban ventilation perspective and the reduction of urban heat islands [36–39].

This study explores the impact of urban winds on a 100 m tall residential building envelope and the effect of the changing microclimate per building height. A critical evaluation is conducted on the design of balconies on the building envelope. The appropriateness of building design according to height is discussed in relation to the direct exposure of balconies to higher-up wind flow speeds, the thermal perception of the residents, and the potential for energy efficiency through a more climate responsive design. Conclusions highlight the need for a holistic understanding of the design of high-rise buildings that goes beyond the pressing issues of energy efficiency. Results form part of a wider research program on the sustainability of the high-rise typology aiming at the successful formation of sustainable design strategies. The objective is a holistic understanding on tall building design by combining issues of energy efficiency and thermal comfort. The studies are based on on-site monitoring of case-studies located in the Mediterranean climate of Tel Aviv.

## 2. Background Studies

The performance of high-rise buildings is influenced both by internal and external conditions. The microclimate of high-rise buildings changes with increasing altitude above ground, more specifically, wind speed increases, while dry bulb temperature drops. Urban

studies have also shown that the opposite is often the case, as the Venturi effect and downward drafts may cause higher speeds at the base of tall buildings [40,41], whereas overshading the pedestrian level may cause significantly lower temperatures there than at the sunny top of a tower [42–44]. The above relationships have a direct effect on heating and cooling loads in relation to height [45,46]. Previous research on the high-rise typology highlighted the issue of increased height in relation to all other construction, and how it affects the structure's energy performance [47–50], the wind regime created at ground level and the surrounding urban fabric; affecting pedestrian thermal perception and movement [51].

Height and arrangement of neighboring buildings play an important role in high-rise construction, where urban wind velocities and solar radiation, or sun shading from surrounding structures may influence their microclimate, thermal behavior, and indoor/outdoor thermal comfort levels [52–54]. While simulations are vital in drawing conclusions on the building's behavior, these are mainly conducted on stand-alone structures, and possibly fail to capture important interconnections and mutual influences with the surrounding urban fabric. Urban studies using on-site monitoring also become important for a more accurate representation of the urban environment, and an understanding of the complexity that governs the design of tall buildings from a variety of angles, i.e., environmental, social, etc.

This paper studies the microclimate that is created along the envelope of high-rise buildings in Tel Aviv, with a special emphasis on wind and gust velocities at different height balconies. The methodology is based on in situ monitoring. Tel Aviv's increasing high-rise building activity over the last few years [55] makes the city a good case for studying the microclimate around such buildings, and assessing the potential for upgrading tall-building design. The city's Planning and Construction Committee issued the 2025 City Master Plan, setting new guidelines for further skyrise development [55], while an increasing number of buildings considerably higher than the existing urban canopy, i.e., 150 m tall and higher, is already massively changing the city's skyline.

Tel Aviv is located in the coastal zone defined as Climatic Zone A in Israel Standard SI 1045 [56]. The climate is characterized by rainless summers and mild, wet winters. Average annual temperatures range around 20 °C, with a minimum average of 13 °C in January, and a maximum average of 27 °C in August. Prevailing wind direction is W and NW, with wind speed averages fluctuating between 3–5 m/s. Global solar radiation can reach 3.43 MJ/m$^2$ in the summer, and 1.53 MJ/m$^2$ in winter. Relative humidity is high throughout the year, with annual averages of 60–67%, so even when summer temperatures are not extreme, there is still heat stress, especially during the months of July and August. The city's long hot season spans from about mid-March to the end of October [57], making cooling and ventilation an integral part of annual indoor acclimatization in buildings in Israel.

Preliminary monitoring studies were conducted in two prominent high-rise office buildings. One is Electra Tower, 165 m tall, and the other one is Midtown Tower, 200 m tall, located in close proximity to the main case-study, the residential tower. Wind and gust velocities were compared between the on-site measurements and the recorded data at the weather station located at the Tel Aviv beach. Gust wind speed is wind speed measured over a smaller period of time, i.e., 2–3 s wind speed, referring to smaller character fluctuations, and represents the maximum gust speed values that people feel the most, while these values are usually double those of wind speed [33,58,59]. Data collection covered two monitoring periods. The first one took place during early-March 2019, this period being considered a transitional winter–spring period with temperatures ranging between 13–18 °C. The second monitoring period was conducted during August (2019) which is considered the heart of the summer. The on-site monitoring stations were located on the rooftops of the high-rise buildings.

Figure 1 depicts the results of the first monitoring period, 6–12 March 2019. Results are given as wind and gust speed averages between the depicted timeframes. Data confirm that gust speeds are usually 2–2.5 times and higher compared to the wind speed. Generally,

gust speeds resemble more the wind speed recordings at the Tel Aviv weather station at the beach, but with higher values at the tower tops. Increases in the wind velocities occur from midday to approximately 21:00 h, while during the night they subside. Gust speed maxima are in the range of 6–7 m/s. On the last day of the monitoring period gust speed averages reached 8.5 m/s, while individual recordings were about 10 m/s. The second monitoring period took place between 7–16 August 2019. Here, again, gust speed averages resemble more closely the recordings at the Tel Aviv beach weather station. In this set of data, one more on-site weather station was located on the balcony of Midtown Tower approximately halfway of its total height, at an approximate height of 100 m, located on the side of the tower, and at an approximate distance of 30 m from its neighboring residential tower (Figure 2, right). Wind and gust velocity averages are in the range of 4–5 m/s, while individual gust recordings at the Midtown balcony reached 8 m/s during 14 August 2019.

The summer recordings presented lower wind velocities (Figure 3). The lowest average recordings are at the balcony level, confirming that wind speed increases with building height. However, gust velocities at the balcony level accelerate to as high as those at the tower tops, and at times reach even higher values. Data indicate that the Venturi effect is at play here, with the narrower cross section of the urban canyon between the towers presenting higher gust velocities than the ones on the roof. A possible explanation for this is that gust velocities are influenced both by the distributions of the average wind speed and the turbulence intensity. As a result, an area with high wind velocities may also have high gust velocities, while an area with low average wind velocities may present high gust velocities if the turbulence is high [60].

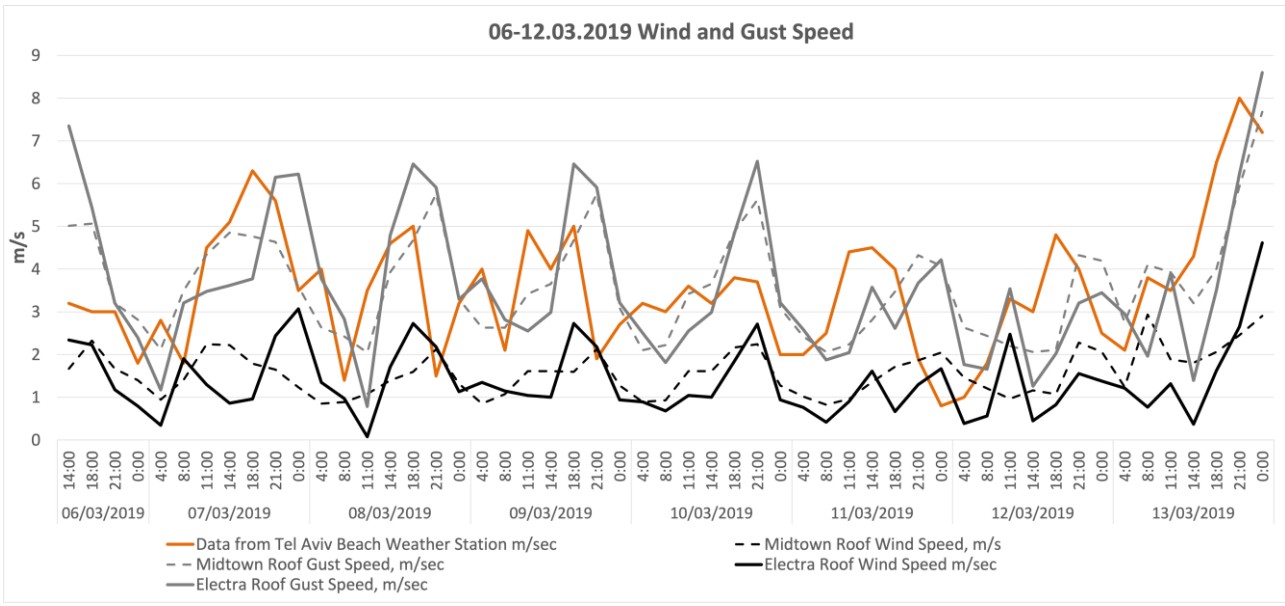

**Figure 1.** Wind and gust speed velocity averages between the depicted timeframes on the building tops of Electra Tower (165 m above ground) and Midtown Tower (200 m above ground). Wind speed recordings from the Tel Aviv weather station located at the beach are also included.

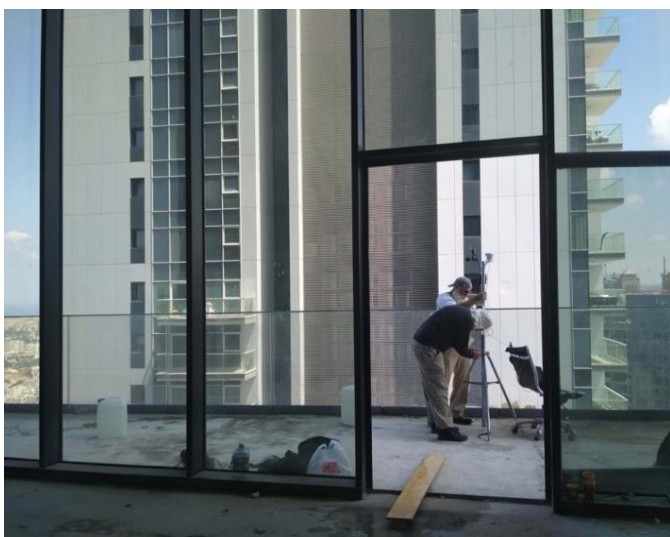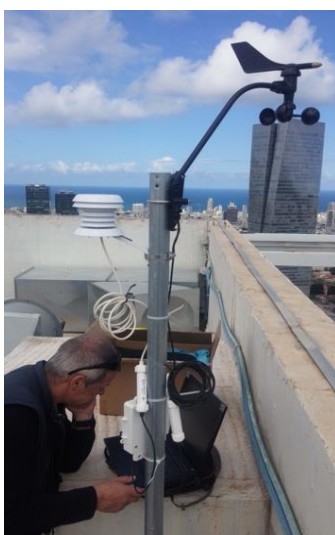

**Figure 2.** (**Left**) view of on-site monitoring station at the balcony of Midtown Tower. (**Right**) view of on-site monitoring station at the roof-top of Midtown Tower, 200 m above ground.

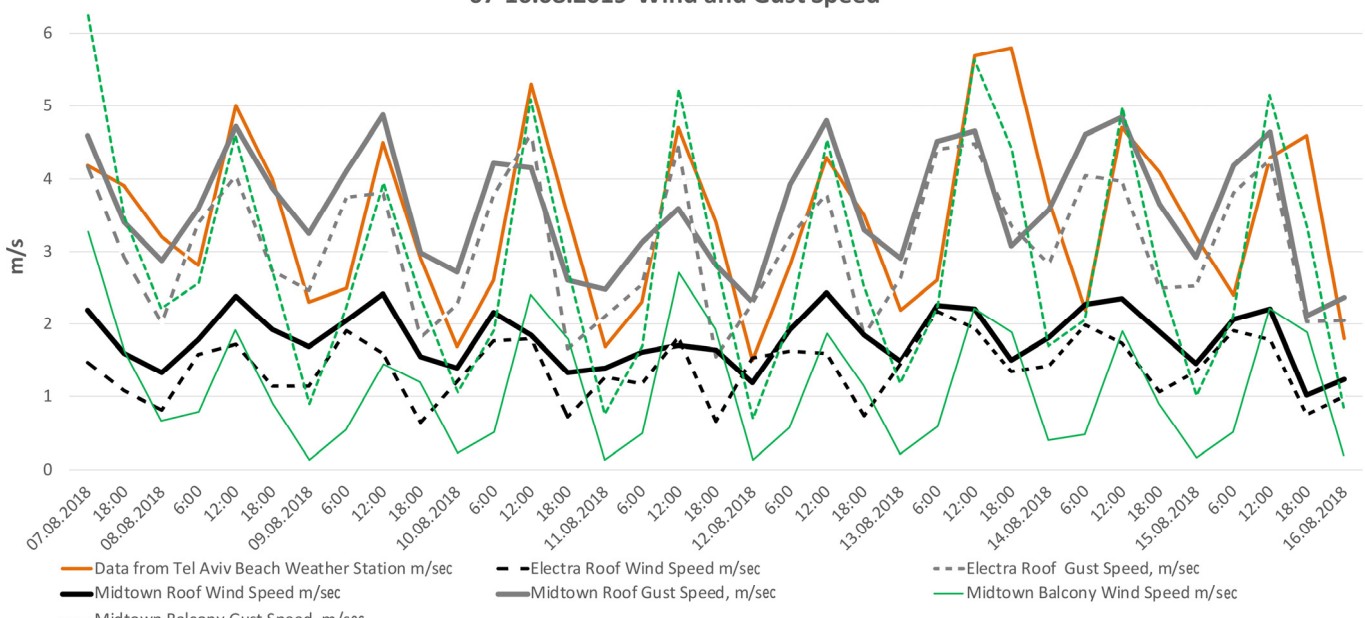

**Figure 3.** Wind and gust speed velocity averages between the depicted timeframes on the building tops of Electra Tower and Midtown Tower, and the balcony of Midtown Tower. Wind speed recordings from the Tel Aviv weather station located at the beach are also included.

The results of these studies prompted further research on the turbulent flow modification of wind and gust velocities per building height in the urban environment based on on-site monitoring. The focus here is on different height balconies and the effect of the wind stratifications per building height of a residential high-rise case study. Conclusions are made on the outdoor thermal comfort of the building occupants (balconies), the safety and usability of the balcony outdoor spaces design at increasing height above ground, as well as regarding missed opportunities for advancing the structure's energy efficiency.

## 3. Materials and Methods

### 3.1. The Case-Study

On-site monitoring of wind and temperature variations is conducted on different height balconies of a residential high-rise case study. The design configuration is considered a typical one, where every floor level is similar to the one above, with no considerations or detailing relating to specific heights above ground, i.e., the design of the balconies along the building envelope. Conclusions are mainly based on a critical evaluation of the residential high-rise building typology in the Mediterranean climate of Tel Aviv (Csa in the Köppen–Geiger climate classification), setting an initial understanding and delineation for a future study's roadmap in similar climates. The study covers a hot and a cold season.

The proposed case study has a West orientation, facing the prevailing W/NW wind direction in Tel Aviv, while it is distanced about 3 km downwind from the coast, with the dense urban fabric of Tel Aviv covering the intermediate space (Figure 4). The city's highway is located nearby, as are the national rail services. The residential tower is a relatively new construction approximately 100 m high and is positioned in close proximity (approximately 15 m) to its twin tower to its south (Figure 5). The towers are located within an urban plot with noticeable high-rise building activity and medium-density 2–3 floor apartment buildings. Below is a height analysis of the case-study balconies. The data are relevant to the CTBUH typical tall building characteristics, in reference to a residential tower, based on the Council's wide database of built projects [61]. The entrance, ground level floor-to-floor height is 4.65 m and a typical floor is 3.1 m high. A half floor is also estimated above the entrance level. Based on the above data, the estimated monitoring height above ground of the balconies, is:

- Floor 4 (West orientation): 4.65 m ground level floor-to-floor height + 3.1 (typical floor height) × 3 + a half floor = 4.65 + 9.3 + 1.6 = 15.55 m
- Floor 15 (South orientation): 4.65 m ground level floor-to-floor height + 3.1 (typical floor height) × 14 + a half floor = 4.65 + 43.4 + 1.6 = 49.65 m
- Floor 17 (West orientation): 4.65 m ground level floor-to-floor height + 3.1 (typical floor height) × 16 + a half floor = 4.65 + 49.6 + 1.6 = 55.85 m
- Floor 28 (West orientation): 4.65 m ground level floor-to-floor height + 3.1 (typical floor height) × 27 + a half floor = 4.65 + 83.7 + 1.6 = 89.95 m

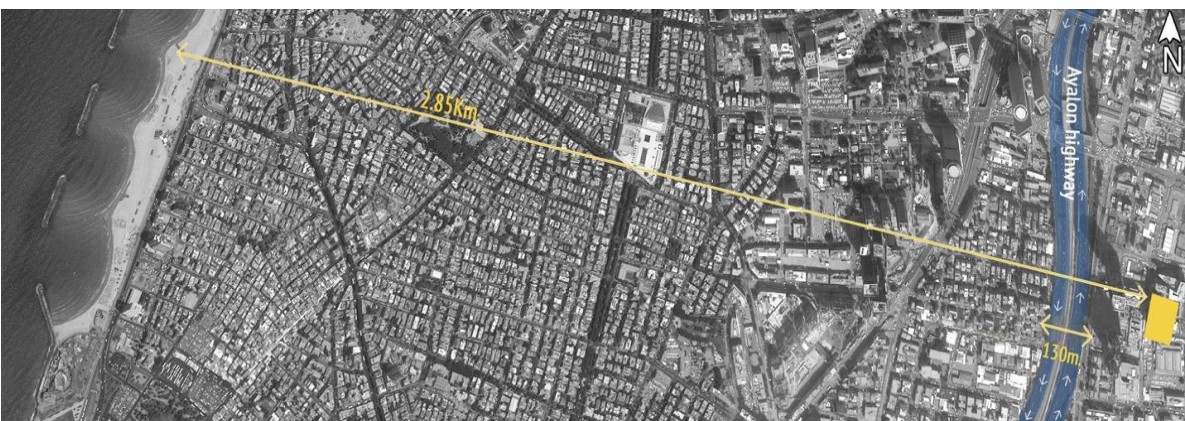

**Figure 4.** Location of residential tower (yellow rectangle (within the urban fabric of Tel Aviv, depicting distance from the sea promenade.

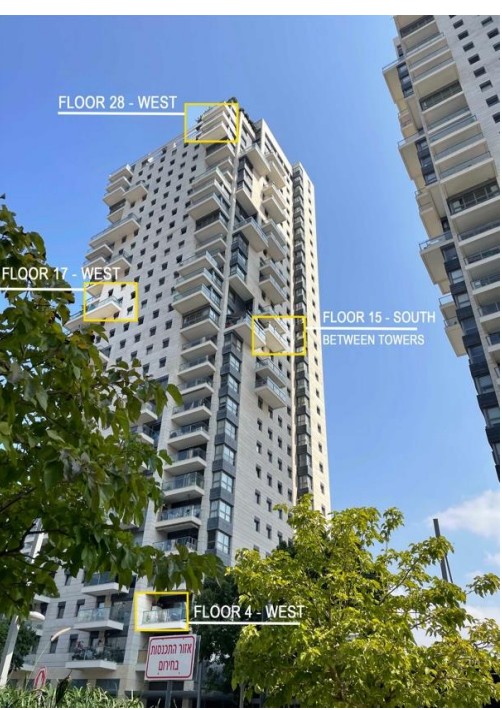

**Figure 5.** View of the residential tower and case-study balconies at floors 4, 17, and 28 on the West windward façade, and floor 15 on the South, side façade—between the two residential towers.

### 3.2. Micro-Meteorological Measurements

In situ micro-meteorological measurements were conducted at different height balconies, representing *summer* and *winter* periods. Monitoring was conducted with HOBO USB Microstations H-21 (Figure 6). Table 1 shows the specifications of the sensors used. Three monitoring locations/balconies were selected on the West windward façade at floors 4, 17, and 28, and one on the South, side façade, between the case-study tower and its neighboring one, both residential, at floor 15 (corner effect) (Figure 5). Floor 15 balcony was chosen in order to better understand the Venturi effect between the two towers, and how it affects the microclimate there. Recorded data included dry bulb temperature (°C), relative humidity (%), wind direction (degrees), wind speed (m/s), and gust speed (m/s).

The *summer* study period spans from 25 June to 1 July 2020, and the *winter* study period from 26 January to 3 February 2021. For the summer period, environmental data were recorded on floor levels 4 and 28 (West orientation—windward façade). For the winter period, initial studies showed considerably higher wind and gust speed velocities that demanded the collection of a larger set of data, as well as a higher level of analysis of the results produced. As a result, monitoring expanded to cover both the windward façades, on floor levels 4, 17, and 28 (West orientation), and the side façade on floor 15 (South orientation—between the towers). Comparisons are made between the summer and winter periods from the in situ meteorological stations, wind velocities from the weather station at the beach, and survey questionnaires administered to the respective residents.

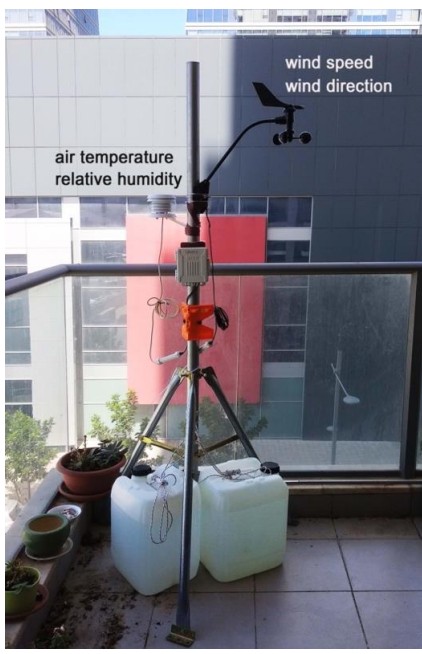

**Figure 6.** Installation of micro-meteorological station for data acquisition: dry bulb temperature (°C), relative humidity (%), wind speed (m/s), gust speed (m/s), and wind direction (in degrees).

**Table 1.** Specifications for HOBO USB Microstation H-21 used for micro-meteorological measurement.

| Parameters | Sensor | Measurement Range | Accuracy | Resolution |
| --- | --- | --- | --- | --- |
| Air temperature (°C) | HOBO S-THB-M002 | −40 °C to 75 °C | ±0.21 °C from 0° to 50 °C | 0.02 °C at 25 °C |
| Relative humidity (%) | HOBO S-THB-M002 | 0–100% | ±2.5% from 10% to 90% | 0.1% RH |
| Wind speed (m/s) | HOBO S-WCF-M003 | 0 to 76 m/s | ±1.1 m/s or ±5% of reading | 0.5 m/s (1.1 mph) |
| Gust speed (m/s) | HOBO S-WCF-M003 | 0 to 76 m/s | ±1.1 m/s or ±5% of reading | 0.5 m/s (1.1 mph) |
| Wind direction (°) | HOBO S-WCF-M003 | 0 to 355° | ±7 degrees | 1 degree |
| Tripod | HSW 2 m Tripod Tower | | | |

Figure 7 shows the micro-meteorological stations in place during the monitoring periods of the respective balconies. The photographs depict quite clearly the urban views toward the sea/west orientation. While there are a number of high-rise buildings in close proximity to the case study, these are not located directly in front of the tower. Except for floor 4 that faces a low-rise building at that level, sited at a distance of more than 15 m, and floor 15 that is located between the residential towers, again at a distance of approximately 15 m, floor 17 and floor 28 are completely exposed. Floor 17 is also exposed from above as there is no balcony directly above it, while floor 28 is shaded and covered by the balcony of the floor above, which is also the last one in the building.

West orientation floor 4.

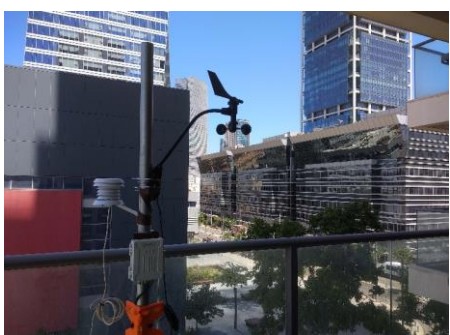

South orientation floor 15

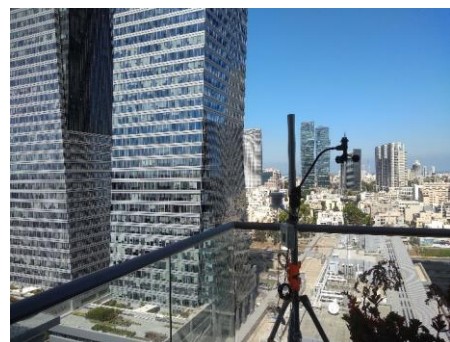

West orientation floor 28.

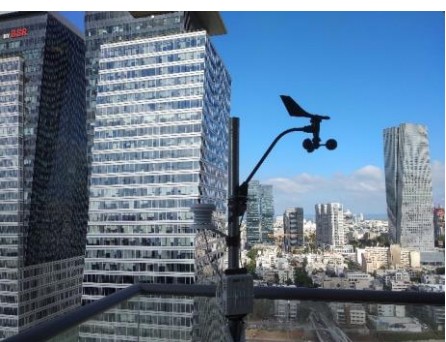

West orientation floor 17.

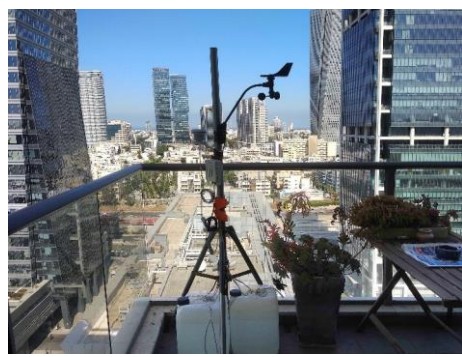

**Figure 7.** Micro-meteorological stations at different height balconies. Top left, clockwise: West orientation floor 4; South orientation floor 15(between the towers); West orientation floor 17; West orientation floor 28.

## 4. Results

### 4.1. Summer Period

Figure 8 graphs depict the summer timeline (25 June to 1 July 2020) in relation to wind and gust velocities, and temperature variations from the micro-mereological stations on floor 4 and floor 28. Wind speed from the weather station at the beach is also depicted as a reference point. The weather station at the beach is located 10 m above ground and is unobstructed by buildings or vegetation, as opposed to the urban location of the residential tower where the monitoring took place. However, the monitored balconies are on average much higher than the surrounding building canopy. The height of the micro-mereological station is 1.5 m, while the case study balcony heights are mentioned further up in the Case-study section. According to the above calculations, the height difference between the weather station at the beach (sea level plus 10 m) and floor 4 is 5.55 m higher, and 79.95 m higher for floor 28. The distance between floor 4 and floor 28 is about 75 m.

Data are depicted as averages between 00:00, 12:00, and 16:00 h. The specific time-frames represent best the daily fluctuations of the environmental variables under study. Monitoring results confirm that wind speed increases with height, while temperature drops. Regarding the temperature, daily values range between 23–29 °C, while the variations between floor 4 and floor 28 are approximately 1–2 °C less for floor 28. In addition, the average diurnal temperature variations are in the range of 1–2 °C less at night, suggesting an overheating of the urban environment. Wind velocities for floor 4 range between 0.3–1.7 m/s, while the increase per building height (floor 4 to floor 28) is in the range of 20% to more than 100%, with values reaching 2–3.4 m/s at floor 28. Similarly, gust speed values at floor 4 range from 1.0–5.0 m/s, while at floor 28 they accelerate to 1.7–7.0 m/s. Gust speed recordings seem to follow more closely the air velocities recorded at the beach weather station. The trend is similar, but with lower values for floor 4, while a better correlation

exists with floor 28, however there, wind spikes can be higher. The highest recorded gust velocity is 7.0 m/s at floor 28, at 16:00 on 26 June 2020, with a temperature of 25.8 °C.

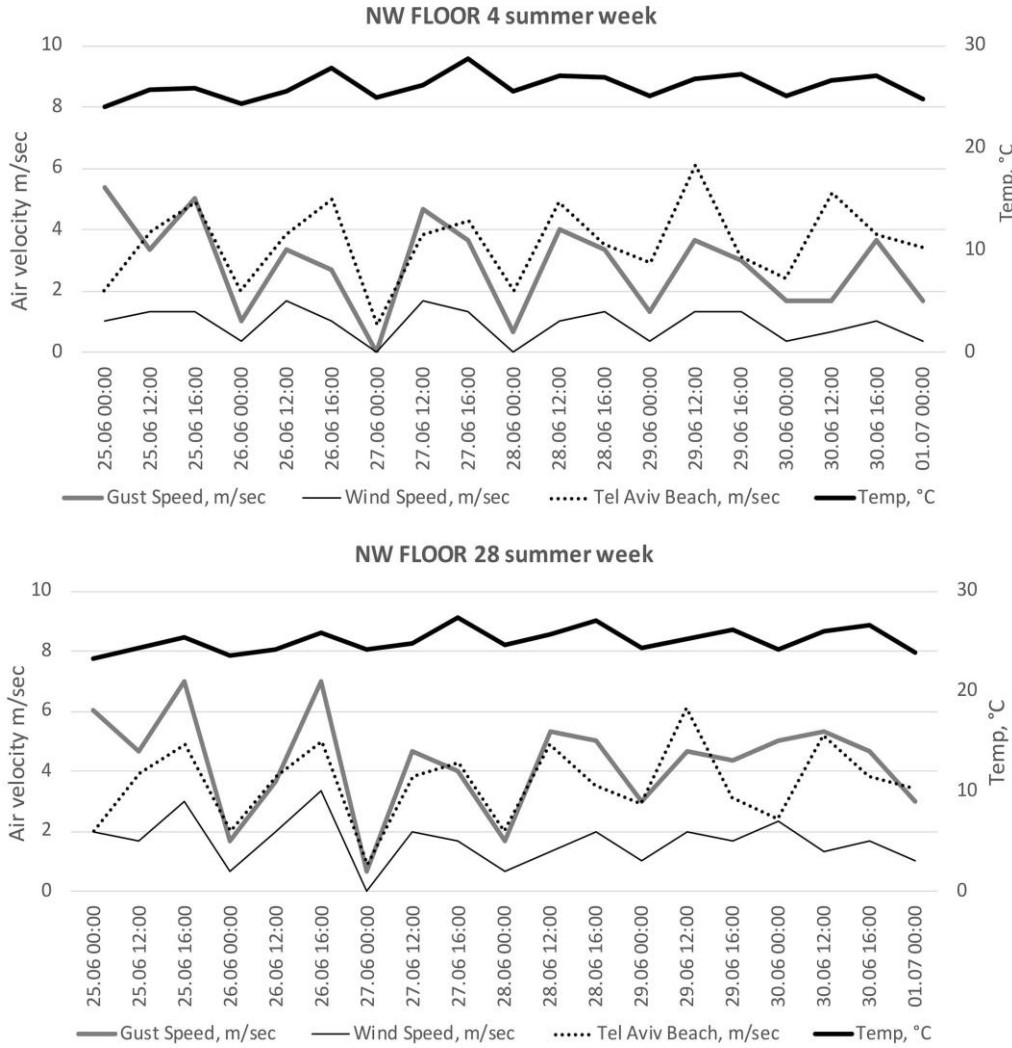

**Figure 8.** Summer period recordings from 25 June to 1 July 2020, depicting wind speed, gust velocity and temperature variations from the micro-meteorological stations located at floor 4 (**top**) and floor 28 (**bottom**) on the West façade, in relation to wind speed from the meteorological station at the beach at a height of 10 m, at 00:00, 12:00, and 16:00 h.

According to Lawson and Penwarden's extensive 'Land Beaufort scale' on wind effects on people at pedestrian level height [62,63], an air velocity value of 7.0 m/s refers to 'force of wind felt on body', and is at the threshold between a 'moderate breeze' and a 'strong breeze'; however, these specifications refer to wind speed at 1.75 m height. Floor 28 is at a height of almost 90 m plus the height of the standing person. In addition, the balcony's frame and railings are made of glass, possibly enhancing a feeling of discomfort to a person standing there, by exposing them to the intensified environmental variables. Figure 9 shows a view of the floor 28 balcony during summer. While the city views are astonishing the balcony is devoid of any decoration due to the high winds. The residents placed a children's pool that is hardly used according to their statements, and that is securely held in place with the weight of a metal chair. The effective use of the balcony space and the opening of the glass doors for ventilation is discussed further on in this paper through an analysis of occupants' questionnaires.

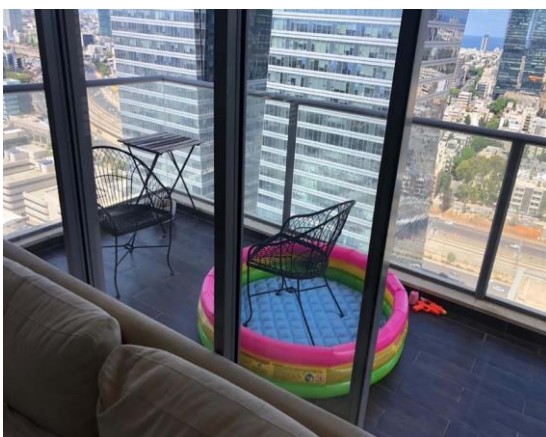

**Figure 9.** View of balcony at floor 28 during summer. The balcony is devoid of decoration due to the high winds, while the children's pool that is seldom used by the owners is secured in place with the weight of the metal chair.

*4.2. Winter Period*

Graphs in Figures 10 and 11 depict the winter period results from 26 January to 3 February 2021, in relation to wind and gust velocities, and temperature variations from the micro-meteorological stations at the residential balconies, as well as air velocities from the weather station at the beach. Due to the higher air flows during winter, the data collection expanded, covering floors 4, 17, and 28 on the windward West façade, and floor 15 on the South, side façade, between the two residential towers. The height differences between the weather station at the beach, located 10 m above ground, and the monitored balconies, are as follows: 5.55 m for floor 4, 39.65 m for floor 15, 45.85 m for floor 17, and 79.95 m for floor 28. Data are depicted as averages between 00:00, 12:00, and 14:00 h. The specific timeframes are considered to best represent the daily fluctuations of the environmental variables under study, while an hourly analysis is also conducted further down in this paper. Daily temperature variations range from about 14 °C at night to as high as 27 °C during the day, with an average of just above 18 °C for the timeframe under study, as recorded at floor 4, 15.55 m above ground. The highest temperatures are observed around 12:00. Monitoring results confirmed that temperature drops with building height between floor 4 and floors 15 and 17; however, there is a slight increase again at floor 28 during winter, e.g., 20.6 °C at 12:00 on 26 January at floor 4, drop to 18.4 °C at floor 17, and rise again to 20.1 °C at floor 28.

Further studies within the urban environment have shown higher temperatures on the roofs of the towers due to their exposure to unobstructed direct and diffuse solar radiation, as opposed to lower levels, certainly the pedestrian level, which are overshaded by the adjoining towers. A thermal comfort study on the specific climatic conditions of Tel Aviv during summers and winters of 2007–2011 reported that the city's all-year thermal comfort range is between 19–25/26 °C [64]. These results resemble the temperature stratifications of the current monitoring data. In hot climates, as per ASHRAE [65], the comfort zone shifts towards warmer conditions compared to colder climates. However, one's perception between comfort and discomfort lies between the calibration of micro-climatic conditions [66]. Outdoor thermal comfort within the urban environment is a multi-faceted subject that involves a number of environmental variables not easily quantified or controlled and is relative to peoples' specific preferences and attributes [67–70]. The impact of urban wind is pivotal. In this paper, the complexity of the outdoor urban environment is discussed in relation to building height through the presence of balconies.

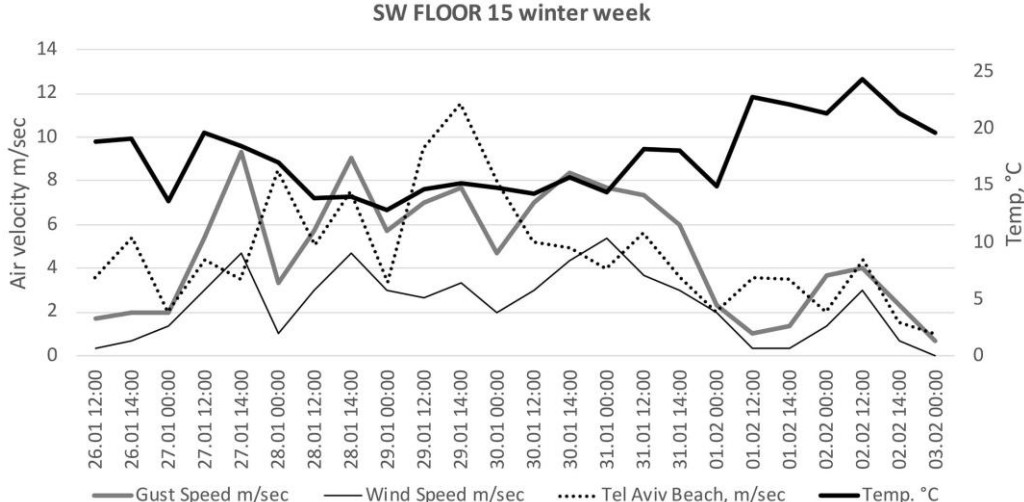

**Figure 10.** Winter period recordings from the 26 January to 3 February 2021, depicting wind speed, gust velocity and temperature variations from the micro-meteorological stations located at floor 15—South, side façade orientation, in relation to wind speed from the meteorological station at the beach at a height of 10 m, at 00:00, 12:00, and 16:00.

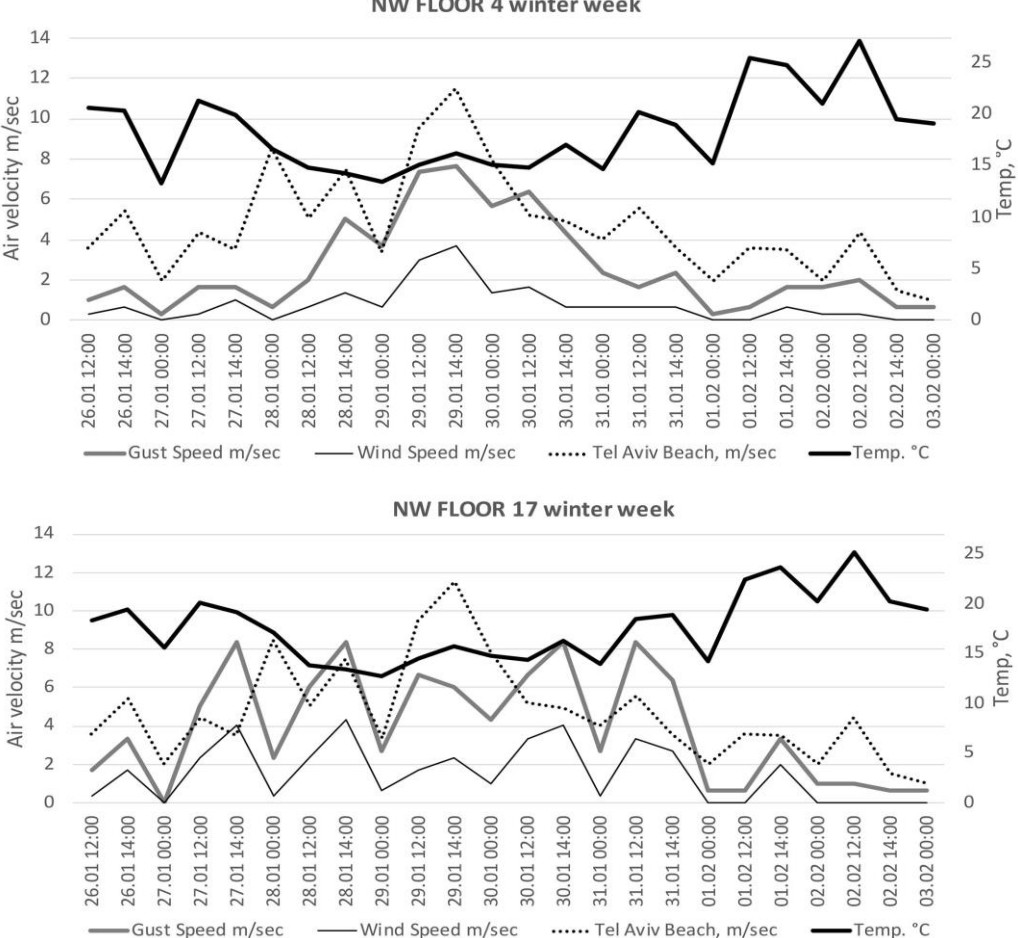

**Figure 11.** *Cont.*

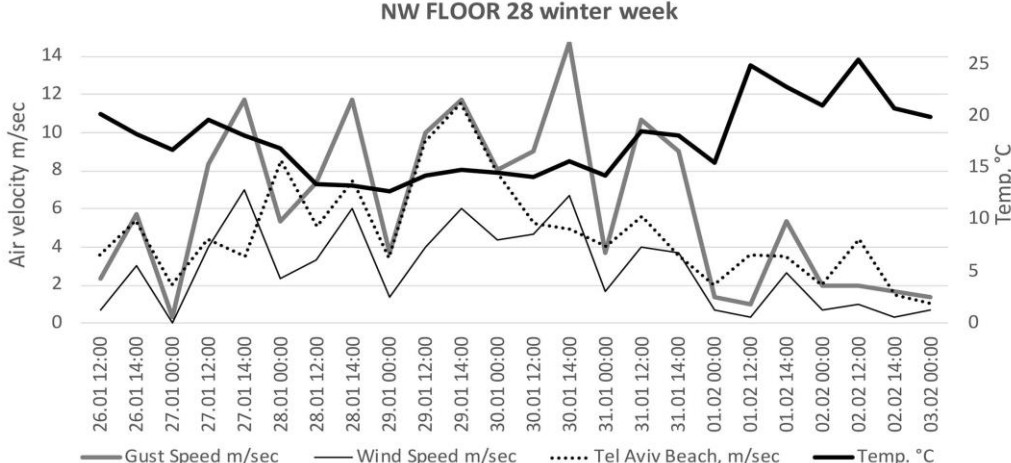

**Figure 11.** Winter period recordings between 26 January to 3 February 2021 depicting wind speed, gust velocity and temperature variations from the micro-meteorological stations located at floor 4 (**top**), floor 17 (**middle**) and floor 28 (**bottom**)—W façade orientation, in relation to wind speed from the meteorological station at the beach at a height of 10 m, at 00:00, 12:00, and 16:00.

During the winter period, monitored air velocities were quite high between 27 January and 2 February, with the beginning and the end of the monitoring week having weak wind days. Results confirm an increase in air velocity per building height. Wind speed averages at floor 4 are in the range of 1 m/s, with a spike to 3.5 m/s on 29 January, at 14:00. The increase, however, from floor 4 to floors 15 and 17, as a general rule, is in the range of 3, 4, and almost 5 times more, e.g., 0.7 m/s at floor 4 rises to 4.0 m/s at floor 17, 4.4 m/s at floor 15 (side elevation—corner effect), and then accelerates to 6.7 m/s at floor 28. Regarding gust speed averages, both summer and winter periods confirm that these usually reach twice the mean wind speed. In addition, similarly with the summer period, monitored gust speed velocities seem to more closely follow the air velocities at the meteorological station at the beach, especially with the increase in building height, e.g., between 29 and 30 January 2021, the trend is exactly the same. Nevertheless, with the increase in building height, both wind and gust speeds relate more to the air velocities at the beach and tend to increase considerably by the time they reach floor 28.

The correlation between air velocities at the beach and the monitoring balconies is especially prominent at the West windward façade, while the trend is not as obvious for the side balcony on floor 15. As a general rule, gust speed increases by approximately double from floor 4 to floors 15 and 17, and then another two times higher up, at floor 28. At 14:00 on 1 January, gust velocity at floor 4 was 4.4 m/s, then increased to 8.4 m/s for floors 15 and 17, and reached 14.7 m/s at floor 28, while at the same time air velocity at the beach was 4.9 m/s. Floor 28 presented considerably higher velocities throughout the windy days that span 27 January to 2 February, with gust speed averages of 8.9 m/s, while at the same time averages for floors 15 and 17 were 5.9 m/s.

In relation to the monitored wind and gust velocities, these are increased twofold and more from the ones during summer. From the graphs of Figure 7, it is quite clear that the velocities of the wind entering Tel Aviv from the beach, as recorded by the meteorological station there, can be quite high, with a maximum of 11.5 m/s on 29 January, at 14:00, and a minimum of 1.5 m/s on 2 February, at 14:00. Cohen et al.'s on-site monitoring study of outdoor pedestrian wind stratifications in three urban locations within Tel Aviv's urban fabric, recorded wind speeds between 0.9–1.9 m/s during winter, and 1.1–2.2 m/s during summer [64]. Similar values were also recorded by these authors during spot measurements, but the scope of these works is not part of this study. Nevertheless, the results highlight the considerable reductions of urban winds from the coastline inland. A further analysis on the diurnal cycle of hourly wind and gust speed velocities in relation to wind direction, and to the air velocities recorded at the beach meteorological station, is

conducted in the following section for a better understanding of the wind behavior per building height during winter.

Diurnal Cycle of Hourly Air Velocities

Figure 12 illustrates a day-and-night hourly analysis on the air velocity stratifications per building height and orientation (West vs. South façade) for the monitored balconies, in relation to the maximum air velocity at the beach during the specific timeframe under study. Results confirm that wind speed increases with height; however, they also underline the importance of wind direction, depicted in degrees from north eastwards on the graphs. During the day, wind accelerates mainly from floor 4 to floor 28; however, in the early morning hours, due to the change in wind direction, floor 17 may present equal or even lower air velocities than floor 4. Nevertheless, it is quite clear than even though floor 4 could also reach high wind velocities, e.g., graph '29.01 p.m.' with daily gust speeds of 8 m/s, wind velocities at higher balconies can extend to the level of danger, making the balconies at higher altitudes dangerous and thus non-usable, redundant. An example is graph '29.01 a.m.', when at 8:30 in the morning an 18 m/s gust speed is recorded at floor 28. While the above example could be considered extreme, the graphs indicate that during high-wind winter days, wind velocities at higher floors range between 4–6 m/s, and gust speeds reach 8–10 m/s on a regular basis at floor 28.

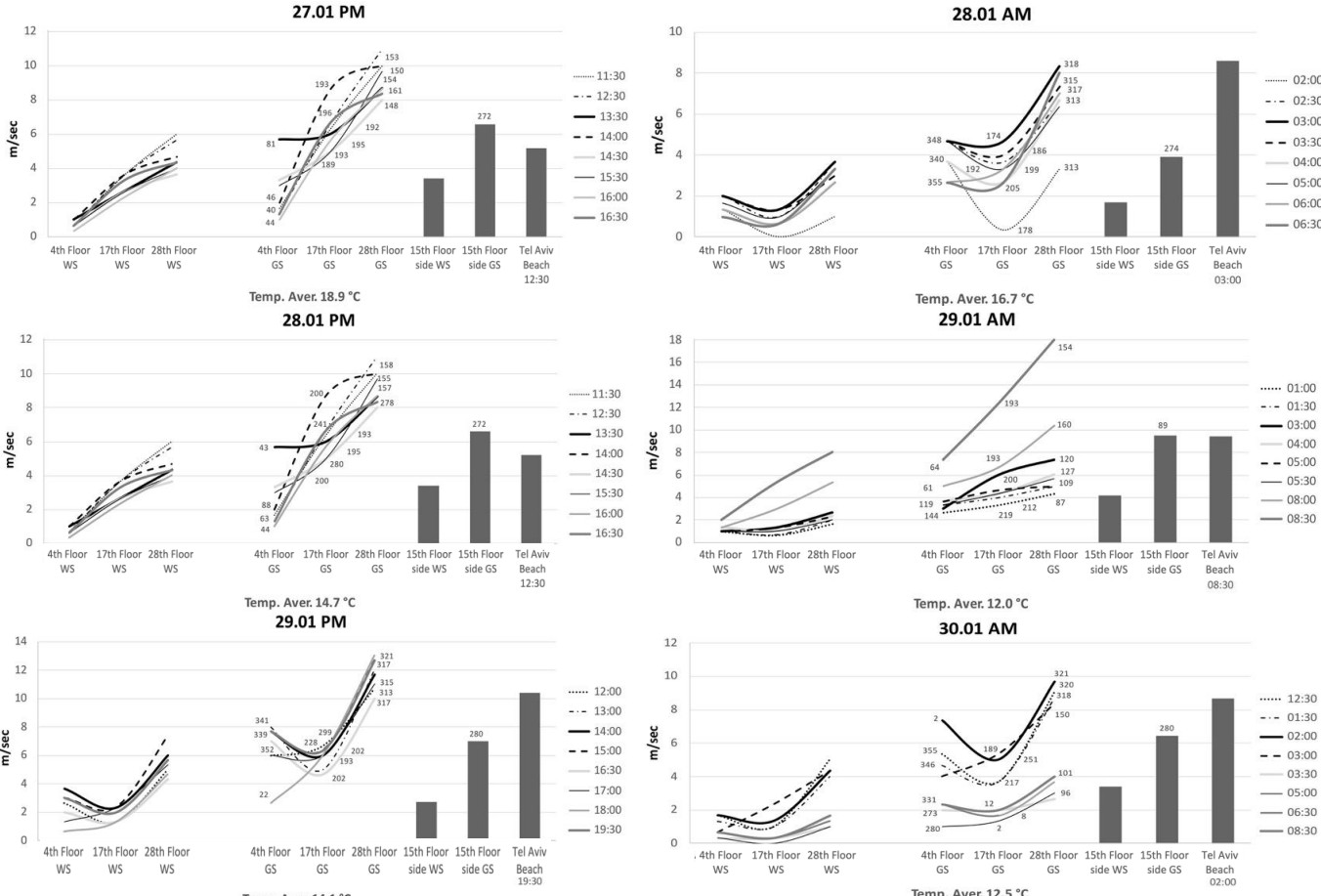

**Figure 12.** *Cont*.

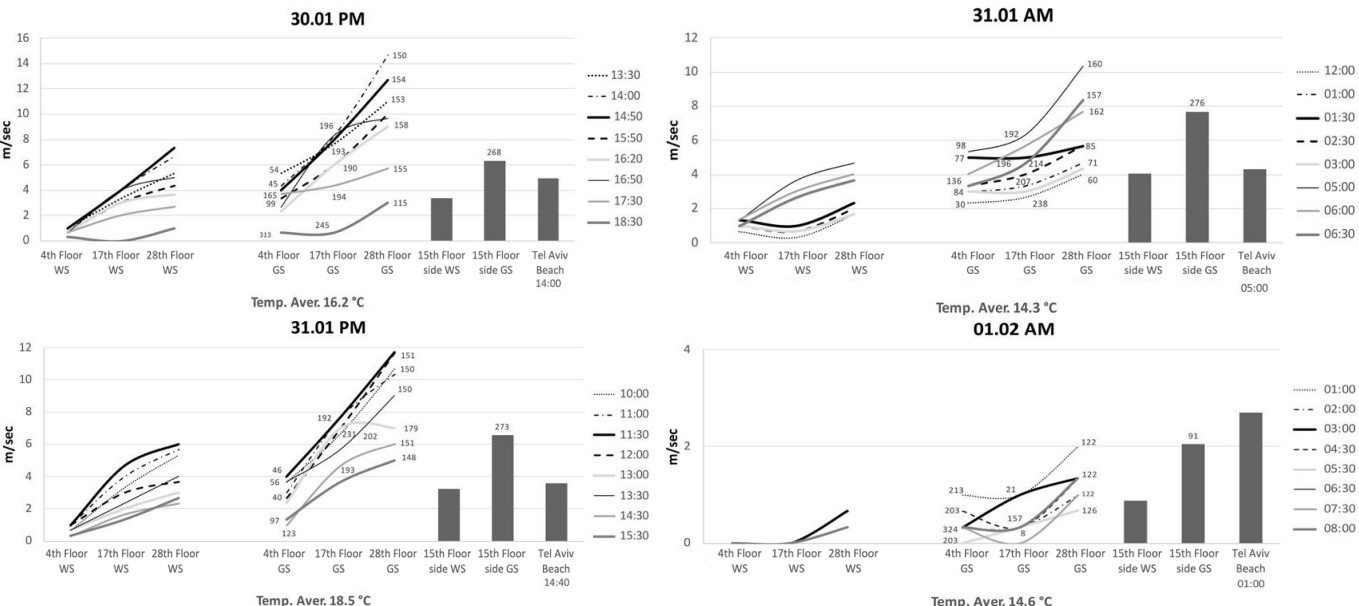

**Figure 12.** Day-and-night maximum wind and gust velocities between 27 January–1 February depicting floor 4, floor 17, and floor 28 (West windward façade) and floor 15 (South side façade-between towers), in relation to the maximum wind velocities at the beach. The numbers indicate wind direction in degrees from N eastwards. The continuous lines of the graph represent the wind flow from lower level to the highest, while the time frames depict the highest values recorded during the specific day.

### *4.3. Wind Comfort Criteria*

Even though this case study focuses on the wind profile per building height, it was important to relate the results of these studies to 'pedestrian height urban winds' and 'wind comfort criteria', in order to provide an understanding on the effects that environmental conditions have on a person occupying the balconies discussed here, and so many others in many similar buildings. However, pedestrian wind comfort is a multifaceted matter that involves a number of parameters. Results go beyond the mechanical effects wind has on people according to the Beaufort scale, i.e., direct effect of wind force [71,72], while the thermal effects or thermal perception describe the physical, physiological and psychological parameters of peoples' comfort conditions outdoors. Physical adaptation relates to physical activity, adding or removing clothing items (reactive adaptation), or an active response to the environment, like the opening of a parasol (interactive adaptation) [73,74]. Physiological adaptation relates to changes that occur after repeated exposure to a stimulus, while psychological factors mainly relate to one's expectations from the environment, and depend on the time of exposure, control and familiarity with the environment, as well as a 'perceived naturalness' [70].

Currently, no universal wind comfort criteria for pedestrians have been agreed upon to date, while existing such are mainly based on research studies (field studies and thermal simulations) [75–77], and may differ amongst different countries and institutions. As a result, more outdoor studies are needed within the urban fabric on a global scale for establishing threshold values between tolerable and unacceptable wind conditions for pedestrians [51,78,79]. Table 2 depicts an evaluation of 'wind comfort and danger' according to Lawson and Penwarden's extended 'Land Beaufort scale' at pedestrian level height [62,63], Lawson's wind assessment in the UK [80], and NEN 8100 wind criteria according to the Netherlands Normalisation Institute (NEN) [81]. The code for pedestrian wind comfort created by the NEN is currently the only official document on a global scale. It is not a legal building requirement, but offers assistance through the provision of strict guidelines for incorporating wind comfort in building construction [72].

**Table 2.** Lawson's assessment in the UK [30], Land Beaufort's scale [Lawson and Penwarden], and NEN 8100 wind criteria.

| The Lawson Wind Comfort Criteria | NEN 8100 Wind Comfort Criteria | Land Beaufort Scale Wind Speed at 1.75 Height | Effect | |
|---|---|---|---|---|
| | | **(m/s)** | | |
| >25 | >20 | 14.6–17.1 | Dangerous to walk People blown over | Strong Gale Poor traversing |
| >20 | | | Unacceptable to walk | |
| 10–12 | 10 | 12.1–14.5 | Fast walking Great difficulty with balance | Near Gale Moderate traversing |
| 8–10 | | 9.8–12 | Pedestrian transit. Inconvenience felt when walking | |
| 6–8 | | 7.6–9.7 | Walking, Sightseeing. Hair blown straight, difficult to walk steadily | Strong Breeze Moderate strolling |
| 6 | 5 | 5.6–7.5 | Short periods of standing/sitting Force of wind felt on body | Gentle Breeze/Moderate Breeze/Fresh Breeze Moderate: Sitting |
| 4 | 2.5 | 3.9–5.5 | Long periods of standing/sitting Raises dust, hair disarranged | |
| | | 2.4 3.8 | Hair disturbed/clothing flaps | |
| <3.9 | <2.5 | 1.1–2.3 | Acceptable for sedentary activities. Wind felt on face | Calm/Light Air/Light Breeze Good: Walking strolling sitting |
| | | 0–1.0 | No noticeable wind | |

Low wind velocity (<1.0 m/s) in hot environments has a negative effect.

Nevertheless, commonly agreed findings between the different indexes see no discomfort for wind speeds up to 5 m/s, some discomfort and unpleasant feelings from 5–10 m/s, and strongly unpleasant feelings and potential danger beyond that [82,83]. However, such data, sparse as they may be, refer to pedestrian comfort and safety, whereas here we discuss comfort and safety on balconies of high-rise residential buildings. This is quite a different discussion since it implies mostly sedentary activities (sitting, dining, etc.), and involves safety discomfort (whether lack of safety is genuine or just perceived) as well as significant hazards, not least in the case of toddlers and the elderly, as well as people with a low body mass. The above monitoring results on air velocities per building height, with a focus on the winter period, range from 'uncomfortable' to 'dangerous', according to the international indexes on wind comfort (Table 2).

*4.4. Resident Survey*

In order to better understand the impact of the intensity of the environmental variables per building height on the thermal perception of the potential balcony users, the subjective human perspective was included in a survey using self-administered semi-structured questionnaires [84,85]. Tables 3 and 4 give an overview of the effective use of the balcony space by their occupants, as an outdoor extension of the apartment area, an understanding of their reliance on the mechanical systems of the building for heating and cooling, and their relationship to natural ventilation, as well as their perception of natural light within the premises. The questions also included the occupants' perception of noise from the highway and railway below, as well as noise from wind gusts.

The responses of the occupants are a combination of their individual preferences on comfort and satisfaction, and the effect of the environmental variables. The responses from floor 4 are the mildest. The balcony is used for recreation throughout the year, infiltration and noises are barely noticed, and natural light levels are pleasant, probably due to this floor level being shaded by a building opposite. The floor 17 balcony has a different character. While the occupants spend up to 1 h per day outdoors, both during winter and summer, they pointed out that this is because they prefer to smoke outdoors and that all balcony

items are tied down due to strong winds, while windows are barely opened for ventilation. Windows and balcony doors also remain mainly closed due to noises from the highway and railway below, while noises due to wind are prominent. During winter, wind and gust velocities are mainly felt in the morning and at dusk, and during summer the sunlight is especially strong at sunset due to the balcony's west orientation.

**Table 3.** Occupant responses on the effective annual use of the balconies A.

| | South—Leeward Façade | West—Windward Façade | | |
|---|---|---|---|---|
| | 15th Floor | 4th Floor | 17th Floor | 28th Floor |
| *Opening balcony glass doors for ventilation* | | | | |
| 1. A few times a day | V | | | |
| 2. Once day | | V | V | |
| 3. Sometimes | | | | |
| 4. Few times | | | | V |
| 5. Not at all | | | | |
| *Perception of light* | | | | |
| 1. Very strong | V | | V | V |
| 2. Strong | | | | |
| 3. Neutral/Pleasant | | V | | |
| 4. Weak | | | | |
| 5. No light | | | | |
| *Is the balcony in use in Winter?* | | | | |
| 1. Daily | V | | V | |
| 2. From time to time | | V | | |
| 3. Not at all | | | | V |
| *Is the balcony in use in Summer?* | | | | |
| 1. Daily | V | V | V | |
| 2. From time to time | | | | |
| 3. Not at all | | | | V |
| *How much time do you spend on the balcony?* | | | | |
| 1. Less than 15 min | V | | | V |
| 2. 15–30 min | | V | | |
| 3. 60 min | | | | |
| 4. More than 60 min | | | V | |

Table 4. Occupant responses on the effective annual use of the balconies B.

| 15th Floor | 4th Floor | 17th Floor | 28th Floor |
|---|---|---|---|
| What is the purpose of the balcony for you? | | | |
| Mainly for storage | Gardening and resting | Resting and hosting | No purpose |
| What objects are on the porch? | | | |
| Baby carrier, equipment (tied down) | Flowerpots, a chair table, a cat litter box | Plants, chairs and table (tied down) | No objects |
| Are there shutters in the apartment? External, internal, both? Are these automatic? | | | |
| Internal, not automatic | No | Internal, not automatic | Automatic external shutters |
| Have you encountered wind gusts, whistles, infiltrations problems, noise? | | | |
| Frequent strong winds. All balcony items need to be secured. Noise and whistle noises are also present. | Barely—only on very stormy evenings in winter. | Windows are not used for ventilation, because of noise and harsh weather phenomena. A lot of noise is present from the highway and train station all day. | Everything—wind, air, noise, whistles. |
| Have you noticed changes between the seasons? | | | |
| Extreme weather during winter | Almost no infiltration. Slightly more in winter when it is stormy. | In the transition seasons strong winds at dusk. | Wind infiltration mainly during winter. |
| Additional comments | | | |
| Prefer natural ventilation over A/C. If not too hot/cold glass doors open and remain open as long as possible. | In the evening the wind is stronger than in the day | Prefer to smoke outside. The most difficult hours are in the summer from noon onwards (western sun), and in the winter the wind gusts in the morning and at dusk. | There is a lot of noise all day from the highway and train station. The windows are mostly closed because it is very cold. |

The occupants on floor 15, on the other hand, prefer natural ventilation to air conditioning (A/C), so even though most of the time wind phenomena can be intense, with gust velocities ranging between 8–12m/s during the day (Table 2), these occupants try to open the glass doors to the balcony for as long as they feel comfortable. However, they do not enjoy staying on the balcony, and only use this space for storing equipment, which they tie down due to the high wind velocities. The perception of light is also strong, and occupants have added Venetian blinds to avoid glare. The most extreme responses are from the occupants of floor 28, that also relate to the pronounced environmental phenomena there. The occupants do not use the balcony at all, except very rarely in summer. They have added outdoor electric blinds and automatic opening sensors, in order to control natural light intensity and introduce some natural ventilation within the premises. Air infiltration due to high wind pressures increases during winter, while whistling noise due to the wind and infiltration, as well as noise from the adjacent highway and railway, are prominent throughout the day.

This section may be divided by subheadings. It should provide a concise and precise description of the experimental results, their interpretation, as well as the experimental conclusions that can be drawn.

The behavioral patterns of the occupants are comparable with the intensity of the recorded environmental variables per building height. Except for floor 4, the character of the other, higher balconies is mainly problematic, and they are not used, thus redundant. On floor 17, the residents' need and will to enjoy their outdoor space overcame their negative feelings stemming from increased wind velocities and noise levels, while on floor 15 the balcony is no different from a storage room. Residents on floor 28 seem to suffer the most and take the least pleasure of this extra space, as they are unable to use it either for pleasure or storage. The higher wind velocities are also counterproductive in terms of natural ventilation and cooling within the premises. These could have been beneficial during the high summer temperatures, while introducing fresh air into the premises, which is an important consideration all year round. In addition, regarding high-rise buildings located near highways and railways, most residents noted that noise levels were quite high, as the sound is not filtered by the surrounding built environment.

## 5. Discussion

This study is part of a wider research on the sustainability of high-rise buildings and their successful integration within the urban environment, given their increasing numbers worldwide. Results emphasize the complications of the varying microclimate in tall building design, and by doing so, emphasize the importance of a different detailing of the building envelope with height, in order to accommodate the changing environmental variables, improve the energy efficiency of the structure, and secure safe and pleasant conditions for the occupants. The building envelope's importance as the interface between indoors and outdoors is paramount [86]. This relationship is relevant to the specific climatic conditions of the building's location, and in high-rise buildings also becomes relevant to the changing microclimate with height [45,46]. Previous studies have confirmed the importance of the thermal properties of the materials of the building envelope on energy loads, with a focus on the high-rise office and residential typologies, based on thermal simulations [49,87,88]. This study draws conclusions on the appropriateness of a typical residential tower design with exposed balconies through on-site monitoring. Monitoring data are juxtaposed with the occupants' perception of comfort in relation to the height of the balconies.

The results in fact revealed that the higher the apartment the less comfortable were its owners to occupy and enjoy the outdoor balcony space (ref. Table 4). These findings stand in contrast with evidence provided of a rent gradient within tall buildings, where rent premiums increase with building height, while the addition of balcony space is considered a further premium [89] the cost of which is both reflected in the price of the apartment and the municipal monthly taxes calculated per floor area. In high-rise Hong Kong, the

provision of a balcony is also considered a 'green feature', as it provides sun-shading, offers plant space, and with height above ground mitigates street pollution and noise, while an apartment without a balcony is expected to sell for considerably less, depending on the surrounding views and location [90]. Conroy et al.'s survey on the subject of 'higher-floor premium' of high-rise buildings in San Diego confirmed a 2.2% increase in sales prices with the addition of every floor level; however, the increase rate decreases with height above ground [91]. Nevertheless, while sale prices per floor level are dependent on a number of variables, e.g., additional building materials with height, size of apartment, provision of car-parking, etc., there is also the perception that units on higher floors are generally more 'desirable' due to extended views, noise and pollution reduction [92–95].

The current studies were conducted in the Mediterranean climate of Tel Aviv, yet it is quite clear, that contrary to the mild climatic conditions, wind and gust velocities tend to intensify considerably with height, reaching the point of danger, especially during winter. Such relationships are especially prominent in the graphs of Figure 12 that depict minimum–maximum winter values between day and night. Regarding the occupants, the main issues that were stressed involved increased air velocities and light intensity, increased infiltration, and wind whistling, as well as noises from the highway and railway at ground level. The intensity of these phenomena relates to the density of the urban environment and the number of high-rise buildings that exist in close proximity to one another. As height increases the structure is exposed both to higher wind and light intensities. Failure to accommodate these into building design results in malfunctioning of the building itself. With regard to the high-rise buildings being located near railroad and railway stations and highways, this is in accordance with the decisions of Tel Aviv's Planning and Building Committee for urban development. However, noise level studies that would provide an understanding on the impact such relationships have on the high-rise occupants are presently lacking.

The current case study rejected the notion of reduced noise from the ground level as height increases, with residents of the higher-up apartments being more aware of noises from the city's highway and railway, than the apartment closer to the street level (ref. Table 4, floor 4). The density of the urban fabric is definitely a parameter here, with the residents of high-rise apartments, in Hong Kong and Singapore for example, where high-rise building density is increased, having a different view on the subject. Three-dimensional simulations of an urban district in Singapore confirm that the changes in noise levels according to height in high-rise buildings are relative to the effects of obstruction, distance attenuation, ground absorption and noise barrier screen [96]. A study on noise level profiles at different levels of three high-rise buildings in Milan confirmed that these are relevant to the characteristics of the urban environment where the specific building is located, and that the results sometimes contravene the general rule that noise levels at higher floors are reduced [97]. As a result, a noise level assessment on high-rise structures becomes important in order to guide mitigation measures.

In terms of design optimization, opportunities are missed for introducing, for instance, a second façade layer enclosing the balcony space. A double-skin façade DSF spanning the width of the balcony could potentially solve the issues noted above, i.e., increased wind and gust velocities, infiltration and noise, as well as improve the effective use of this outdoor space at higher altitudes and advance the energy efficiency of the structure. The detailing of the DSF façade, however, becomes very important [88,98]. The exterior layer of glass doors could have the possibility of opening and closing subject to the use patterns of the occupants and the changes in environmental variables, while considerably reducing annual energy loads [49,88]. The balcony's outdoor space could act as a greenhouse during winter, warming passively the interior, while during summer the strategic opening of the glass doors alongside dynamic shading could allow for the introduction of natural ventilation within the premises, irrespective of the floor height above ground, reducing cooling loads. However, examples of opening doors—partitions may be problematic, even at the ground floor level, let alone in high-rise buildings. Better yet, new technologies could make such

relationships easier to accomplish. The above strategy could result in annual energy load reductions and a more sociable and friendly outdoor space at greater heights.

The introduction of a double-skin façade layer into the high-rise balcony space could be explored further, while other strategies could also be appropriate in order to accommodate more socially acceptable, energy-efficient design proposals in the high-rise building typology [99–101]. Nevertheless, similar monitoring studies on high-rise buildings are lacking, while the inclusion of the occupants' perception through surveys appears to be vital towards an understanding of indoor/outdoor human comfort conditions in tall buildings. Additionally, further research, to micro-model air flow conditions using powerful CFD tools, could also prove valuable in the design of high-rise buildings. The goals and results of this study underline the complexity of the high-rise typology and the need for further studies on the relationship between design and function, especially in relation to the changing microclimate with height.

## 6. Conclusions

The growing numbers of high-rise buildings globally, a result of urbanization and population growth, highlight the need for in-depth studies relating to their economic, environmental, and social characteristics. The outcome will be the implementation of design strategies relating to the tall building typology, in order for it to successfully contribute towards the establishment of sustainable urban environments. This case study questions the appropriateness of the design of exposed balconies along the high-rise building envelope in the Mediterranean climate of Tel Aviv. On-site monitoring at different height balconies recorded the changing environmental variables with height and the increased wind flow fields around the tower. The results were juxtaposed with the perception of comfort of the respective occupants.

The conclusions of this case study highlight that while there are different parameters governing tall buildings relative to their height above ground, such as environmental, social, economic, etc., current design practices do not reflect upon such, with the use of a typical floorplan design all along the structure's height being one example. The introduction of exposed balconies along the high-rise building envelope, a design solution which is currently gaining popularity, especially in warmer climates, does not seem to yield the success expected. The balconies' copy/paste approach from one floor level to the next is not appropriate as height increases, and may also reach a level of danger like, for example, the environmental conditions faced in this case study by the residents of floor 28 during winter.

The tall building typology has seen a dramatic evolution from its birth to date. The main design parameters relate to function, architectural style, height and structural strategy, and accordingly affect the structure's energy performance over time [102–104]. The new generation of tall buildings, however, from the lower-height range to the tallest, will have to address issues of energy efficiency, an increasing level of climate change mitigation strategies, as well as the provision of a safe and comfortable environment for the people occupying them. Planning authorities should become stricter and require a higher level of analysis of the proposed designs, e.g., a detailed environmental impact assessment report addressing the building's impact on its surroundings, as well as the impact of the changing climate with height. An ideal scenario would be the establishment of green design guidelines on high-rise development according to climate, as well as incentives for 'excellence' in design, as part of the planning procedure.

In addition, tall buildings and their relationship to the urban environment should be studied from a holistic perspective and that can only be achieved through the partnership and commitment of all parties involved in city planning, urban and building design. In this process, a successful dialogue between government agencies, policy makers, urban designers, city planners, academics, architects and engineers, and their clients, becomes vital and should form a continuous process of information and expertise exchange. Such procedures should become the norm towards a sustainable urban future, where high-rise buildings could be seen as assets, instead of problematic urban additions.

**Author Contributions:** Conceptualization, ideas, formulation, and evolution of research goals and aims took place by all the authors present in this paper. Preparation, creation (writing the initial draft) and presentation of the published work conducted by I.A.M. Critical review, commentary or revision by I.A.M. and H.I.-B.-S. Visualization/data presentation by S.S. with the assistance of H.I.-B.-S. All the results/experiments and other research outputs were conducted by S.S. and H.I.-B.-S. Oversight and leadership responsibility for the research activity planning and execution, including mentorship was by I.A.M. Instrumentation and other analysis tools were supplied by I.A.M. and H.I.-B.-S. Experiments and data/evidence collection were completed by S.S. and H.I.-B.-S. Statistical, mathematical, computational, and other formal techniques to analyze or synthesize study data were analyzed by S.S. All authors have read and agreed to the published version of the manuscript.

**Funding:** This research received no external funding.

**Data Availability Statement:** Data available on request from the authors.

**Acknowledgments:** This research is partly supported by the Department of Civil and Environmental Engineering, Faculty of Engineering Sciences, Ben-Gurion University of the Negev (BGU); and BGU's Center for Energy and Sustainability. Access to Midtown Project and Electra Tower was facilitated by the buildings' CEOs and managers, I. Hamamy and M. Bar Lev, respectively, and their staff. We kindly acknowledge their cooperation and kind assistance, as well as those of the residential tower's residents whose balconies were studied.

**Conflicts of Interest:** The authors declare no conflict of interest.

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
