# Peer review of "Climatic Variability in Altitude: Architecture, Thermal Comfort, and Safety along the Facade of a Residential Tower in the Mediterranean Climate"

_buildings, doi:10.3390/buildings13081979_

Round 1
Reviewer 1 Report
An excellent paper: it discusses a topic that is of great importance, but has so far met little interest amongst researchers.
The analysis of the meteorological data is thorough and sound, based on the comparison of the weather stations and the in-situ measurements.
I found the comfort evaluation of the residents very interesting, since it highlights an aspect that is often overlooked in the assessment of conditions on the balconies of high-rise buildings.
The discussion section is a pleasure to read. Two points that might be worth considering:
a) the double skin facade has been proven to be not cost effective in similar climate conditions (BESTFACADE project, 2007)
b) I am not really sure that the balconies could be 'transformed' to attached greenhouses. In the past the use of doors that open has been proven to be not very practical, even at ground floor level, let alone in high rise buildings. (Papadopoulos, 1996) Perhaps technology has made things easier for the user, but openable windows still take a lot of space, need to be secured and need an 'active' user
Finally, it would be interesting, as further research, to micro-model air flow conditions using a powerful CFD tool
Author Response
Thank you very much for your appreciation of our project, your comments and suggestions.
a) A new sentence and reference has been added. 'The detailing of the DSF façade, however, becomes very important (Christian et al., 2008; Saroglou et al., 2019)'.
b) the comment has been taken under consideration and added to the paper. Unfortunately, the authors were not able to recover the publication (Papadopoulos, 1996).
c) your comment has been taken under consideration and added to the paper.
Reviewer 2 Report
This paper discusses a timely subject of intertwined impact of architecture, thermal comfort, and safety along the facade of a tall buildings.
The paper is informative and significant but needs some structural modifications to be published:
Abstract: Please mention the key findings in the abstract. At the current form it is more like a proposal abstract. The abstract needs to be able to stand alone. This is essential for future referencing and citation.
Materials and Methods: Background studies needs to have its own dedicated section (will be section 2). Materials and methods should be section 3 starting with the case study.
Line 236 Highway is one word!
Line 246: above the sea level? or above the street/ground level? why do we need to know the sea level altitude here?
Figure 12: the data on different floors may not be continuous. When presenting them with lines arranged by floors on the X Axes they may come misleading. Also the titles of each graph and the time scale of individual graphs. Some start from 11:30 and others from 12:00, 1:00, 1:30! I think it is better to present them with a time scale on the X Axes and use multiple lines for the wind speed at different heights (4th, 17th, 28th floors).
Line 521-532: some text overlaps in formatting - this may be only a printing issue.
Please use subtitles for the discussion section. The discussion section needs some links to the results. In its current form, it is relevant but detached from previous sections.
Author Response
Thank you very much for your comments . They have been taken under consideration.
a) the occupants perception is included in the abstract.
b) Background studies are now section 2 , and Materials and Methods section 3.
c) the word has been amended
d) has been amended to say 'above ground'
e) In Figure 12 the continuous lines of the graph represent the wind flow from lower level to the highest , while the time frames are not the same because the aim it to depict the highest values recorded during the specific day , could differ from day-to-day. This is also clarified in the caption.
f) tables and figures and been mentioned in the Discussion section , connecting the results with the comments made
Reviewer 3 Report
In the article measures and analyses of the wind speed values on different floors of high-rise buildings were considered. The measuring equipment was placed on balconies on different floors. Measurements were carried out in summer and winter. It has been estimated that the wind speed increases with increasing altitude. This is quite well-known information. However, the authors also conducted an assessment of subjective feelings regarding the use of balconies by residents of different floors.
It turns out that the wind speed on different floors affects the way the balconies are used. Too high wind speeds mean that users cannot use balconies. The analysis also took into account the fact that it is common for apartments on higher floors to be more expensive, and the presence of a balcony is an advantage. It turns out that balconies on the upper floors can be problematic. The authors also noted the noise. The authors formulated conclusions resulting from the analyses and suggested some examples for improvements.
The article has the correct structure and is written correctly and understandably. There may be a lack of emphasis on research limitations.
In addition, please review all text to standardize the units. The unit for speed appears as m/sec (line 133) and m/s (line 204). Please check it in the entire text. Additionally, attention should be paid to indices in units, e.g. MJ/m2 (line 134). From line 520 there was bad formatting that made it impossible to read part of the text.
Author Response
Thank you very much for the good review and the comments made. They have been taken into consideration.